# The Cardiovascular Phenotype in Fabry Disease: New Findings in the Research Field

**DOI:** 10.3390/ijms22031331

**Published:** 2021-01-29

**Authors:** Daniela Sorriento, Guido Iaccarino

**Affiliations:** 1Department of Advanced Biomedical Sciences, Federico II University, Via Pansini 5, 80131 Naples, Italy; daniela.sorriento@unina.it; 2Research Center on Hypertension and Related Conditions CIRIAPA, Federico II University, Via Pansini 5, 80131 Naples, Italy

**Keywords:** fabry, lysosomal disorder, cardiovascular disease, inflammation, mitochondrial dysfunction

## Abstract

Fabry disease (FD) is a lysosomal storage disorder, depending on defects in alpha-galactosidase A (GAL) activity. At the clinical level, FD shows a high phenotype variability. Among them, cardiovascular dysfunction is often recurrent or, in some cases, is the sole symptom (cardiac variant) representing the leading cause of death in Fabry patients. The existing therapies, besides specific symptomatic treatments, are mainly based on the restoration of GAL activity. Indeed, mutations of the galactosidase alpha gene (GLA) cause a reduction or lack of GAL activity leading to globotriaosylceramide (Gb3) accumulation in several organs. However, several other mechanisms are involved in FD’s development and progression that could become useful targets for therapeutics. This review discusses FD’s cardiovascular phenotype and the last findings on molecular mechanisms that accelerate cardiac cell damage.

## 1. Introduction

Lysosomal storage disorders comprise a group of rare inherited metabolic diseases characterized by abnormal deposition of intracellular wasting materials due to enzyme deficiencies. Among them, Fabry disease (FD) is an X-linked inherited disorder of glycosphingolipid metabolism due to deficient or absent lysosomal alpha-galactosidase A (GAL) activity, which results in a progressive accumulation of globotriaosylceramide (Gb3) and its metabolites [1,2]. Such a mechanism is considered responsible for the damage of several organs, including kidney, heart, lung, small intestine, brain, and liver, thus allowing FD classification as a “multiorgan” disorder. The diagnostic signature is the detection of Gb3 in urine and plasma, that is confirmed by genetic analysis. In some FD patients, especially females, urine Gb3 are not altered, limiting its use as a diagnostic tool [3,4]. Recently, lyso-Gb3, a deacylated analogue of Gb3, emerged as a novel indicator of FD, showing a greater sensitivity than Gb3 in FD females [4]. Although Fabry patients share the same pathogenetic mechanism (defect in GAL activity and accumulation of Gb3 in tissues), FD appears a “multifaced” disorder, and patients show significant variability in clinical signs. According to the canonic classification, Fabry phenotypes categorize into two specific groups classic and non-classic. The classic phenotype includes the manifestation of multiple symptoms during childhood or adolescence with males affected earlier than females. The main symptoms comprise neurological pain, acroparesthesia, and episodic “Fabry crises” of acute pain as well as significant renal, cardiac, and cerebrovascular complications that could manifest in later stages. In the non-classic phenotype, the clinical signs have mainly a later onset (fourth to the sixth decade of life), but they could manifest in childhood with very different symptoms from those of the classic phenotype. Adult-onset cardiac (cardiomegaly, left ventricular hypertrophy, cardiomyopathy, hypertrophic cardiomyopathy, and myocardial infarction) and renal (end-stage renal disease) variants are more prevalent. Although FD is an X-linked disorder, heterozygous females have symptoms ranging from very mild to severe due to random X-chromosome inactivation [5]. Thus, FD is a very complex condition due to the significant variability of clinical phenotypes. Research in the field is still ongoing to better understand the pathogenesis and identify new molecular targets. In this review, we deal with the cardiovascular phenotype of Fabry patients, which represents the most frequent cause of death, and the involved molecular mechanisms.

## 2. The Experimental Models of Fabry Disease

Finding a cure for diseases involves a long journey from bench to bedside, starting with extensive preclinical study in vitro (cell culture) and in vivo (animal models of pathology) and ending with clinical trials to test the efficacy of proposed drugs. This route is even harder for FD, given the lack of a good animal models reproducing the pathology’s key phenotypes. Thus, cells from Fabry patients seem to be the only available effective model to study the molecular mechanisms involved in the development of FD (fibroblasts, [6,7,8] and peripheral blood mononuclear cells (PBMC) [9,10]. Recently, new cellular models of Fabry disease have been generated to study the molecular alterations that precede the cascade of events leading to cardiac alterations. Induced pluripotent stem cells (iPSCs) line, INSAi002-A, were generated from skin fibroblasts from a hemizygous FD patient with a rare nonsense mutation, p.W287X [11]. However, studying the cardiac cell is more relevant to evaluate cardiac alterations at the molecular level in the early stages. To this aim, human iPSC-derived cardiomyocyte (CM) from patients with FD carrying mutations in GLA (c.458G > A; c.658C > T) were generated [12]. These cells show FD’s key features, including accumulation of Gb3, increased excitability, with altered electrophysiology and calcium handling [12]. Proteomic study in these cells revealed the accumulation of several proteins in the cardiomyocyte proteome (LIMP-2) and secretome (cathepsin F and HSPA2/HSP70-2), which is recovered by genetic correction [11]. These data suggest the potential of this cellular model to study the molecular mechanisms underpinning FD, allow a first screening of effective drugs and identify new cardiac biomarkers for FD. However, the cellular model lacks a physiological relevance since it is a simplified, high controlled system that cannot reproduce the interplays between cells, tissues, and organs. To date, two murine models of FD have been generated, GAL-KO and Tg/KO mice, which reproduce Gb3 accumulation in tissues and organs. The GAL-KO is a mouse bearing a deletion of the endogenous galactosidase alpha gene (GLA) [13]. In this model, Gb3 accumulates in several tissues such as kidneys, heart, and liver, but it cannot induce organ damage. The lack of typical clinical signs of FD prevents the use of this preclinical model for therapeutic purposes. However, this model has been useful to evaluate the effects of the enzyme replacement therapy (ERT) [14] and gene therapy [15,16] on restoring GAL activity. A second mouse model of FD (Tg/KO mice) has been proposed, lacking endogenous GLA but expressing a human R301Q GLA transgene transcriptionally regulated by the human GLA promoter. The R301Q mutation is common in Fabry patients with both classic and late-onset symptoms [17]. Patients carrying the R301Q mutation developed hypertrophy of both the ventricular septum and the left ventricle. There is a significant age-dependent accumulation of Gb3 in disease-relevant tissues in these mice [18], but the presence of tissue and organ damage is still unknown. Studies are in progress to evaluate Tg/KO phenotype and generate novel effective animal models of FD. Figure 1 summarizes the available experimental models of FD and Table 1 summarizes their features.

## 3. Pathogenetic Mechanisms

Gb3 accumulation in tissues and organs due to the GAL enzyme’s inactivation represents the common pathogenetic mechanism in FD. Over 600 mutations of GLA gene have been identified until now (missense or nonsense point mutations, splicing mutations, deletions, or insertions) associated with a reduced or null function of GLA enzyme [19]. Depending on mutations, the complete loss of GAL enzymatic activity is generally associated with severe and early-onset Fabry phenotypes. A significant reduction of such activity is related to mild late-onset phenotypes [20]. Some GLA gene mutations seem to be neutral variants or potential modifiers rather than triggers of the disease, such as the A143T variant [21]. In this context, the International Fabry Disease Genotype-Phenotype Database has been generated to collect information on gene mutations that cause the two major clinical subtypes of Fabry disease (available at http://dbfgp.org/dbFgp/fabry/FabryGP.html). This database is continuously updated based on new findings from the literature. Recently, a novel GLA mutation, c.270C>G (p.Cys90Trp), was identified in a Lithuanian family with a classical form of Fabry disease in heterozygous women with predominant cardiac phenotype [22]. Furthermore, a novel severe mutation, a gross deletion of 3′ region of the GLA gene including coding parts of exon 7, has been highlighted in a 55 years old woman with the diagnoses of hypertrophic cardiomyopathy (HCM), high blood pressure and dyslipidaemia [23]. The GLA gene mutation c.337T>C (p.F113L) is associated with a late-onset phenotype of FD with predominant cardiac manifestations [24]. Additionally, p.N215S, p.M296I, p.R301Q, p.G328R, and IVS4+919G>A mutations are associated with a later-onset cardiac phenotype of Fabry disease [25]. Nevertheless, it is difficult to establish an exact correlation between genotype and phenotype in Fabry patients due to phenotypic variability among individuals with the same genotypic variant, including intra-familial variability [26]. Inter- and intra-familial variation in phenotype could also, at least in part, depend on other genetic modifiers, environmental factors, and epigenetics [26,27,28]. This evidence suggests that defective GLA activity cannot be considered the sole cause of the clinical manifestation, and other mechanisms should be involved. In this context, other hallmarks of FD have been identified, such as endothelial and mitochondrial dysfunction, and inflammation. These phenotypes are involved in the disease’s development or progression and represent the primary regulators of cardiovascular dysfunction in FD. 

### 3.1. Endothelial Dysfunction

In Fabry patients, an increased intima-media thickness and an impaired artery flow-mediated dilatation are associated with plasma lyso-Gb3 level independently from age and sex, suggesting an early onset of atherosclerosis [29]. Such a clinical picture correlates with elevated serum levels of MMP-9 and angiostatin, which indicate an increased extracellular matrix turnover in Fabry patients [30]. Furthermore, the increase of VEGF-A serum levels, a specific endothelial cell mitogen, suggests vascular damage [31]. Thus, endothelial dysfunction is a feature of FD. At the molecular level, endothelial dysfunction depends on the excessive production of reactive oxygen species (ROS). In endothelial cells from a forearm skin biopsy of Fabry patients, the accumulation of Gb3 is associated with increased production of ROS and increased expression of cell adhesion molecules [32]. Accordingly, in human cardiac endothelial cells, Gb3 accumulation alters essential endothelial proteins (eNOS, iNOS, COX-1, and COX-2). However, this effect disappears in response to GLA silencing [33] suggesting that endothelial dysfunction is mainly due to the intracellular accumulation of Gb3 rather than to defects of GAL activity [33]. The involvement of endothelial Gb3 accumulation in the development of vasculopathy in Fabry patients is still controversial since the removal of stored glycosphingolipid from the endothelial cells does not always prevent the progression of vascular disease [34]. Thus, further studies are needed to clarify this issue better.

### 3.2. Impairment of Mitochondrial Quality Control

Mitochondrial quality control is an essential process for cell health, allowing elimination and replacement of damaged mitochondria. Lysosomes exert a crucial role in removing damaged mitochondria by mitophagy, thus suggesting that mitochondrial function defects could be present in lysosome storage disorders. Mitochondrial function and metabolism have not been extensively investigated in Fabry pathology even if the few available findings suggest an alteration of autophagic flux and energy metabolism [7,10,35,36]. However, such an issue should be further explored, focusing on other mitochondrial mechanisms, such as mitochondrial dynamics, which are essential for the mitochondrial network quality control, and could be altered in FD.

#### Alterations of Autophagy

Ivanova and colleagues showed that the lysosomal abnormalities in FD also cause impairment of autophagy. Indeed, in blood mononuclear cells from Fabry patients, Beclin1 and LAMP1 levels increase and, consistently, LC3-II and its binding complex, SQSTM1/p62 are reduced [7]. The same authors also show that mitochondrial function is affected in these cells. Indeed, ATP synthesis is impaired, probably due to the disruption of the mTOR pathway, the mitochondrial membrane potential is reduced, and cytochrome c levels are increased. Autophagy and mitochondrial impairments are both, in part, restored by enzymatic replacement therapy (ERT), suggesting that these phenotypes strictly depend on GAL activity [7]. These observations find confirmation in other tissues and cells from Fabry patients. Electron microscopy of renal biopsies from Fabry patients shows an accumulation of vacuoles that decreases with ERT. The altered LC3 and p62/SQSTM1 in these tissues support autophagic flux impairment [8]. A similar phenotype occurs in cultured fibroblasts isolated from cutaneous biopsies of Fabry patients [8]. Moreover, in a human podocyte model of Fabry’s disease, the accumulation of intracellular Gb3 increases autophagosomes [37]. Accordingly, LC3-II levels are increased, and mTOR kinase activity is reduced [37]. All these findings suggest that altered autophagic flux an expected phenotype in FD; in this context, autophagy of specific organelles (i.e., mitophagy) needs to be better explored.

### 3.3. Mitochondrial Dysfunction

Many lysosomes storage disorders present mitochondrial dysfunction, including reducing mitochondrial membrane potential, ATP content, and increased ROS production. The impairment of autophagic flux, in turn, further worsen this damage [38]. FD fits well in this context since, besides alterations of autophagy, mitochondrial dysfunction also occurs. In fibroblasts from Fabry patients, respiratory chain enzymes activity and energy metabolism, are significantly reduced [9,10]. Such impairment of oxidative phosphorylation could be due to alterations of membrane lipid composition [39], as it occurs in cardiomyocytes differentiated from GLA-KO human embryonic stem cell line [40]. This hypothesis is in line with previous findings demonstrating that glycosphingolipids bind strongly to mitochondrial membranes and markedly alter mitochondrial function and ATP production [41]. Damaged mitochondria, in turn, could increase ROS production. Besides fibroblasts, mitochondrial function is also impaired in PBMC from Fabry patients probably due to alterations of mTOR signalling [7]. Based on such findings, we can speculate that mitochondria play a double role in FD being both background actors and active players. Further studies are needed to confirm this hypothesis and open new possibilities in the treatment of FD. 

### 3.4. The Inflammatory Phenotype

Lysosomes have a significant impact in many processes, including immune responses. Indeed, these organelles are involved in antigen presentation and processing, phagocytosis, and release of pro-inflammatory proteins [42,43]. Therefore, defects in lysosome function, as in FD, could affect the immune system. This hypothesis is supported by the proof of the concept that chronic inflammation is a hallmark of FD. Gb3 accumulation in lysosomes can activate immunological processes that progress to a chronic inflammatory state due to the stimulus’s continuous presence [44]. The infiltration of inflammatory cells, such as macrophages and lymphocytes, is detected in several organs of Fabry patients and is likely associated with key impaired processes such as autophagy, apoptosis, and oxidative stress [44].

Furthermore, serum levels of IL-6, IL-1β, TNF-α, monocyte chemoattractant protein-1 (MCP-1), intercellular adhesion molecule-1, and soluble vascular adhesion molecule are significantly higher in Fabry patients compared with controls [44]. Inflammatory cytokines and regulators were also increased in cultured PBMC from Fabry patients [6] and in tissues from Fabry knockout mice [45]. Inflammation is, therefore, a common feature in FD, and it also contributes to worsening pathology progression. Some inflammatory cytokines, such as TGF-β1, which are already known to accelerate nephropathy [46], are highly expressed in kidneys from Fabry mice, suggesting their involvement in organ damage progression. Gb3 accumulation promotes fibrotic remodelling of the arterial wall at the vascular level, resulting in a shear stress-dependent increase of pro-inflammatory factors, such as NFkappaB, worsening endothelial dysfunction [34]. 

## 4. The Cardiac Phenotype 

Cardiovascular diseases are the leading cause of death in Fabry patients; left ventricular hypertrophy (LVH) and myocardial fibrosis are the main risk factors for death [20]. LVH is the most common clinical sign shown by echocardiography in several Fabry patients [47]. Hypertrophy starts with concentric remodelling, which progressively evolves to overt hypertrophy with cardiac fibrosis and reduced contractile performance [48]. However, hypertrophy could be not revealed in the early stages of the disease. At the same time, echocardiography could detect diastolic dysfunction [49], suggesting that alterations of diastolic functions could precede cardiac hypertrophy. Diastolic dysfunction is a common feature of Fabry disease while left ventricular systolic dysfunction reduces ejection fraction or fractional shortening rarely occurs [48]. A high frequency of ischemic events and myocardial infarctions is also detected in Fabry patients. A reduced myocardial perfusion reserve characterizes Fabry disease while the peripheral artery endothelial function is preserved [50]. Electrophysiological abnormalities and arrhythmias could also occur [51,52,53]. The most frequent rhythm abnormalities include supraventricular tachycardias and atrial fibrillation and flutter. Valvular structural abnormalities are frequent, especially in the left heart valves, due to valvular infiltration [54]. Advances in the knowledge of FD’s cardiac phenotype are due to cardiac magnetic resonance (CMR). CMR allows a non-invasive tissue characterization, including the assessment of myocardial fibrosis, through late gadolinium enhancement, and sphingolipid storage, by T1 mapping. A study from Nordin and colleagues clarified the developmental stages of cardiac phenotype in FD patients based on these parameters [55]. This phenotype’s evolution starts with an accumulation phase that occurs in childhood and is characterized by ECG changes and low T1. Myocyte hypertrophy and inflammation phase follows with chronic troponin increase, low T1, and left ventricle posterior wall thinning, especially in women. The latest step is characterized by persistent LVH and troponin increase, fibrosis, NT-proBNP elevation, and clinical heart failure [55].

The cardiovascular phenotype in FD is not shown in all patients and, when it occurs, appears with variable severity. A possible explanation could be different types of mutations since some seem to be more strictly associated with cardiac disease. Ile239Met mutation in the GLA gene occurs in a family with a predominant cardiac phenotype of Fabry disease [56]. Individuals from this family carrying Ile239Met mutation display LVH and were mainly females [56]. Furthermore, the p.F113L mutation of the GLA gene seems to cause the late-onset phenotype of FD with predominant cardiac manifestations [24]. However, multiple mutations could occur in the same individual, which allow the development of LVH. Thus, the involvement of other pathogenetic mechanisms in cardiac dysfunction development, as described below, should also be considered.

### 4.1. The Cardiac Variant 

There are some rare cases, especially in male hemizygotes, in which cardiac hypertrophy is the only overt clinical sign of FD [57]. These patients do not show the typical symptoms of FD but develop heart failure. These cases are defined as “cardiac variants” and are characterized by a Gb3 accumulation exclusively in the heart, causing cardiac damage without affecting the other organs. Several findings suggest that this phenomenon is probably due to residual alpha-galactosidase A activity, which delays the disease’s progression, thus preventing the early manifestation of clinical signs. [58]. Since this phenomenon depends on alpha-galactosidase A activity, it could be associated with specific GLA gene mutations.

Even if the cardiac variant occurs mostly in male hemizygotes, a rare case of a woman with FD’s cardiac variant has also been described. The genetic test revealed a missense heterozygous mutation in the GLA gene, c. 395 G/G>G/A, p. G132E, and the urinary sediment showed Mulberry cells’ presence [59]. These data suggest screening all patients with heart failure without typical FD symptoms for urinary Mulberry cells to allow early diagnosis and prompt therapy [59]. 

### 4.2. Molecular Mechanisms of Cardiac Dysfunction

How does Gb3 accumulation in the heart alter cardiac function? Several reports propose the involvement of different molecular mechanisms. CM differentiated from a patient-derived iPSC show accumulation of Gb3, GAL activity deficiency, and ANP expression increase [60]. These features are associated with increased excitability, impaired electrophysiology, and altered calcium handling [61] due to increased expression of the membrane Ca2 + sensor L-type calcium channels, hyperphosphorylation ryanodine receptor and decreased expression of phospholamban [61,62]. Such findings suggest that defects in calcium handling could contribute to developing the cardiac phenotype in FD. Cardiac energy metabolism was also shown to be altered in FD [35]. In GLA-null cardiomyocytes, Gb3 accumulation associates with impaired autophagic flux, alteration of mitochondrial membrane potential, and increased ROS production [40,63,64]. These evidence suggest the crosstalk between loss of GAL activity and mitochondria dysfunction in cardiac cells. Given these findings and the known role of mitochondria in the development of heart diseases [65,66,67,68], it is likely to suppose that mitochondrial alterations could play a primary role in the development and progression of the cardiac phenotype in Fabry patients.

Inflammation is also associated with cardiac dysfunction in FD. Indeed, a multicenter cohort study in Fabry and control patients revealed that inflammatory and cardiac remodelling biomarkers are elevated in Fabry patients and correlate with a faster disease progression in patients with more severe clinical features [69]. Furthermore, a clinical study shows that myocarditis is detectable in half of Fabry patients with cardiomyopathy and correlates with disease severity [70]. Giving the known role of inflammation in the development of cardiovascular diseases [71,72], it is believable that the chronic inflammatory state could trigger or worsen cardiac damage in Fabry patients. In this context, anti-inflammatory therapy could become adjuvant to prevent irreversible cardiac damage in FD.

In multiple degenerative and acute diseases, mitochondrial dysfunction and inflammation are strictly linked [73]. Indeed, mitochondrial dysfunction increases ROS production, causing oxidative damage in both the heart and the vasculature, and therefore, inducing vascular inflammation. ROS triggers the NFkappaB pathway leading to activation of NLRP3 inflammasome. The generation of inflammatory cytokines, in turn, causes endothelial and mitochondrial dysfunction by further activating NFkappaB [73]. Furthermore, the activation of NFkappaB is also involved in alterations of endothelial function and expression of pro-hypertrophic genes [74,75]. Thus, the hypothesis raises that a vicious circle is established in response to Gb3 accumulation, mainly based on NFkappaB activation as an amplifier of processes disruption and cardiac cell damage in FD (Figure 2). If such a hypothesis is confirmed, NFkappaB could be a novel target in FD. 

## 5. Available and Potential Therapeutic Interventions

The approved therapies for the treatment of Fabry patients mainly focus on recovering GAL activity, which is the main, but not the sole, pathogenetic mechanism of FD. The ERT was approved in Europe in 2001. Two enzyme preparations of the defective AGAL are commercially available, agalsidase alpha (Replagal, Shire) and beta (Fabrazyme, Genzyme) administered intravenously. ERT effectively improves patient outcomes and delays disease progression [76] but comes with the counterpart that patients poorly tolerate the intravenous infusion. Moreover, ERT has a limited tissue penetration, cannot cross the blood-brain barrier, and induces anti-drug antibodies’ production. The IV infusion of the recombinant GAL proteins triggers an immune response, requiring the combination with neutralizing anti-drug antibodies (ADAs) [77]. An alternative therapy has been proposed by Amicus Therapeutics, oral chaperone therapy with Migalastat (Galafold), approved in Europe in 2016 [78]. GLA gene mutations resulting in a reduction of enzymatic activity due to protein misfolding are the required criterium for Migalastat treatment. Indeed, this small molecule chaperone acts by stabilizing endogenous GLA enzyme and allows a proper folding of the protein in the endoplasmic reticulum. The therapy with Migalastat is sufficient to reduce cardiac mass compared to ERT. This oral therapy eliminates the requirement for lifelong intravenous infusions and is well tolerated since Migalastat is a non-immunogenic molecule. This small molecule has enhanced cell and tissue penetration and can cross the blood-brain barrier [79]. However, this treatment has several limitations. A specific mutation is required for enrollment in a migalastat therapeutic regimen, meaning that just an elite of Fabry patients (35%–50%) could benefit from this drug [80]. Chaperone reduces LVH, but its effects on renal function is still to clarify. Indeed, long-term treatment with migalastat was associated with generally stable renal function [81] even if several factors could affect this response, such as basing proteinuria levels. Moreover, blood pressure control could be a discriminant in reducing the estimated glomerular filtration rate [82]. Thus, further investigations are needed on this issue. Several clinical trials show that the treatment with migalastat for 12 months results in a significant decrease of the left ventricular mass index, but plasma lyso-GB3 levels are not affected [82,83]. Patients who switched from ERT to Migalastat present this feature, suggesting that combined therapy is not entirely effective.

Of course, patients are also subject to symptomatic treatments: Diphenylhydantoin and carbamazepine for pain; dialysis for renal failure; a pacemaker for cardiac arrhythmias. 

It is clear that an efficient, well-tolerated drug for all Fabry patients does not exist yet and advancements in this research field are needed to increase the quality of life of Fabry patients and their life expectancy.

Based on recent discoveries on this condition’s pathogenesis, it is likely to suppose that inflammatory and mitochondrial phenotypes could be promising for therapeutic purposes. Thus, targeting inflammatory pathways could be a useful adjuvant strategy in managing FD and the use of autophagy/mitophagy inducers to improve mitochondrial function. NFkappaB, which represents a common pathologic factor between inflammation and energetic failure, could be tested as a therapeutic target. These phenotypes deserve to be better explored. 

Given the increasing availability of therapeutic options, in the next future it will be important, also, to investigate the impact of treatments on the FD phenotypes, based on the causative mutation of the disease. This could be achieved by means of investigation of available data in the literature, by a metanalysis including studies in which the genetic characterization of the mutation(s) responsible of the diseases and the analysis of the effects of the intervention have been investigated at the same time. Moreover, future studies can be performed in which different interventions are compared in patients harboring same causative mutations. The hallmarks of FD, including available interventions, are summarized in Figure 3.

## 6. Conclusions

FD is a multifaceted condition with variable clinic phenotypes among patients. The cardiovascular one is the most recurrent and is also the primary cause of death. The main trigger of the dangerous machine is the lack of GAL activity and the accumulation of Gb3 in lysosomes, which are, in fact, the main target of available therapies. However, in the last decade, other important cardiovascular mechanisms have been revealed, such as genetic mutations, endothelial dysfunction, alterations of energetic metabolism, and activation of inflammatory molecules. Among them, inflammation and mitochondrial dysfunction seem to be the most promising possible target, sharand NFkappaB is a potential common mechanism. The lack of experimental models makes difficult the investigation of the molecular mechanisms of FD. Most studies are performed in human cells derived from patients. Among them, iPSCs derived from Fabry patients carrying specific mutations could be a useful model for diagnostics and therapeutics, within the scope of personalized medicine based on genotyping. However, the mouse model is fundamental for translational purpose, and to date, no mouse model reproducing FD’s main clinical features is available yet. Thus, it is essential to generate an adequate FD animal model for testing specific treatments and increase the knowledge about the pathogenetic mechanisms. This will allow developing more effective drugs to save and facilitate all patients affected by FD.

## Figures and Tables

**Figure 1 ijms-22-01331-f001:**
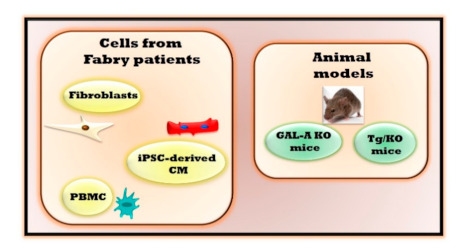
Fabry experimental models. The study of Fabry disease is performed in both cellular and animal models. Cells which are isolated from Fabry patients include fibroblasts, induced pluripotent stem cells (iPSCs)-derived cardiomyocyte (CM) and peripheral blood mononuclear cells (PBMC). Two in vivo animal models are available for studying the physiopathology of Fabry disease: Mice bearing a deletion of the endogenous galactosidase alpha gene (GAL-A KO) and mice lacking endogenous galactosidase alpha gene (GLA) but expressing a human R301Q GLA transgene transcriptionally regulated by the human GLA promoter (Tg/KO mice).

**Figure 2 ijms-22-01331-f002:**
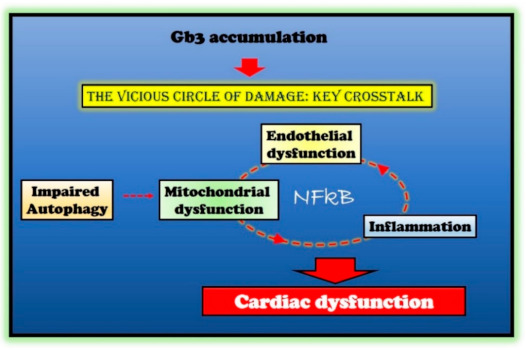
Molecular mechanisms in Fabry disease (FD) cardiac damage. Globotriaosylceramide (Gb3) accumulation in Fabry disease impairs several critical processes within the cell (autophagy and endothelial and mitochondrial function) and induces an inflammatory state. The crosstalk between these impaired conditions is responsible for cardiac damage in Fabry patients. The nuclear factor kappa-light-chain-enhancer of activated B cells (NFκB) represents the critical node in this vicious circle of damage.

**Figure 3 ijms-22-01331-f003:**
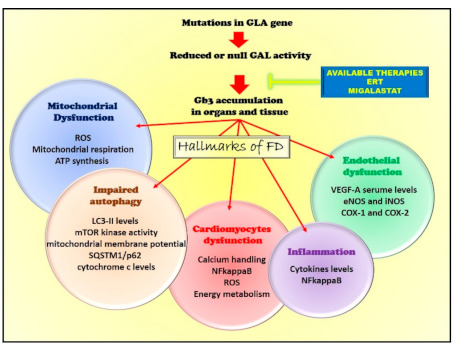
The hallmarks of FD. Mutations in GLA impairs alpha-galactosidase A (GAL) activity leading to Gb3 accumulations in organs and tissues. This causes several cellular alterations leading to the development of the key pathologic features of Fabry disease (endothelial dysfunction, inflammation, cardiac dysfunction, and alterations of autophagy and mitochondrial function). The available therapies, enzymatic replacement therapy (ERT) and Migalastat, target GAL activity to restore the enzyme’s optimal levels and function.

**Table 1 ijms-22-01331-t001:** Main features of Fabry experimental models.

**In Vivo Experimental Models**	**Main Features**	**Advantages**	**Limitations**
Fibroblasts derived from Fabry patients	Alterations of autophagy [6], mitochondrial respiration and energy metabolism [7,8]	Optimal model to evaluate molecular mechanisms involved in cardiac fibrosis.	The in vitro model lacks a physiological relevance preventing the evaluations of different cells crosstalk that could interfere in the pathogenetic mechanism.
Induced pluripotent stem cells -derived cardiomyocytes *(iPSC-derived CM*)	Accumulation of Gb3 [12] Increased excitability with altered electrophysiology and calcium handling [12], accumulation of LIMP-2 cathepsin F and HSPA2/HSP70-2 [11]	Optimal model for evaluating cardiac alterations, cardiac safety and efficacy for evolving drugs.	
Peripheral blood mononuclear cells (*PBMC*)	Impaired mitochondrial function [10], inflammatory cytokines and regulators [9]	Alterations in PBMC can reflect the same ones in the heart	
**In Vivo Experimental Models**	**Main Features**	**Advantages**	**Limitations**
Mouse with deletion of endogenous galactosidase alpha gene *(GAL-A KO)*	Gb3 accumulation in kidneys, heart, and liver [13]	Optimal model to evaluate the effects of drugs on restoring GAL activity	No organ damage
Mouse lacking endogenous GLA but expressing a human R301Q GLA transgene transcriptionally regulated by the human GLA promoter *(Tg/KO)*	Age-dependent Gb3 accumulation in disease-relevant tissues [18]	No data are available yet to define the effectiveness and advantages of this experimental model	This model could reproduce only the clinical signs of Fabry patients which are associated with R301Q specific mutation.

## Data Availability

All Data generated or analyzed are contained within the present article.

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
