# Peer review of "The Cardiovascular Phenotype in Fabry Disease: New Findings in the Research Field"

_ijms, 2021, doi:10.3390/ijms22031331_

Round 1

Reviewer 1 Report

Figure 3: The hallmarks of FD : In my opinion, this figure could be more defined

 2. The experimental models of Fabry disease : a table could be added to summarize the existent experimental models with their strength and weakness

Author Response

Figure 3: The hallmarks of FD: In my opinion, this figure could be more defined

REPLY: We replaced the figure 3.

  1. The experimental models of Fabry disease: a table could be added to summarize the existent experimental models with their strength and weakness

REPLY: We are grateful to the Reviewer for suggestion and added the table in the text (pages 3-4).

Reviewer 2 Report

The manuscript entitled “The cardiovascular phenotype in Fabry disease: new findings in the research field” by Sorriento and Iaccarino, is a review concerning with Fabry disease, a lysosomal storage disease. Among the LSDs, Fabry disease is one of the most studied and about 200 reviews have been published in the last couple of years (according to Pubmed). The present review is specifically committed to the description of the cardiac variant and in particular, it highlights the molecular aspects of the disease.

Two disease-specific therapeutic approaches are already available: the enzymatic replacement therapy and the pharmacological chaperone therapy (migalastat).  However, more research is required since both therapies have some side effects and limitations of application.

In this context, I think the review deserves to be published.

In the present form, it is clearly written.

In order to provide a more comprehensive description at molecular level, I would suggest adding, if possible, a list of the mutations mainly responsible of the cardiac variant.  Moreover, I would suggest highlighting the best therapeutic approach for these mutations (according to the literature data available). This is a very important issue, since the responsiveness to the therapy is strictly related to the specific mutation.  Overall, these details could help for a better stratification of patients and finally choose the most appropriate therapy.

Author Response

The manuscript entitled “The cardiovascular phenotype in Fabry disease: new findings in the research field” by Sorriento and Iaccarino, is a review concerning with Fabry disease, a lysosomal storage disease. Among the LSDs, Fabry disease is one of the most studied and about 200 reviews have been published in the last couple of years (according to Pubmed). The present review is specifically committed to the description of the cardiac variant and in particular, it highlights the molecular aspects of the disease.

Two disease-specific therapeutic approaches are already available: the enzymatic replacement therapy and the pharmacological chaperone therapy (migalastat).  However, more research is required since both therapies have some side effects and limitations of application.

In this context, I think the review deserves to be published.

In the present form, it is clearly written.

In order to provide a more comprehensive description at molecular level, I would suggest adding, if possible, a list of the mutations mainly responsible of the cardiac variant.  Moreover, I would suggest highlighting the best therapeutic approach for these mutations (according to the literature data available). This is a very important issue, since the responsiveness to the therapy is strictly related to the specific mutation.  Overall, these details could help for a better stratification of patients and finally choose the most appropriate therapy.

  • REPLY: We are grateful to the Reviewer for the suggestions, we modified the manuscript accordingly. We discussed in the text the main GLA mutations which are strongly linked to late-onset phenotype with heart involvement (lanes 131-134). We added in the potential therapeutic approach the need for further study to evaluate the association between genetic mutation and sensitivity to treatments, as suggested by this Reviewer. To tackle this task, a proper metanalysis should be performed, including studies in which the genetic characterization of the mutation(s) responsible of the diseases and the analysis of the effects of the intervention have been investigated at the same time. We now discuss it in the paper (lanes 403-409).

Reviewer 3 Report

The cardiovascular phenotype in Fabry disease: new findings in the research field.

Sorriento D, Iaccarino G.

Int J Mol Sci 2021 …

Sorriento D, et al. wrote a comprehensive review about Fabry disease cardiovascular phenotype. In a well-written and interesting/complete paper, they analyze the mechanisms of cell damage. I have only minor comments:

Since the paper do not include abbreviations section, figures 1 to 3 need a legend (pages 3, 7, and 9, respectively).

Page 1: space between groups and classic (line 39)

Page 2: cells (line 64)

Page 5: ventricular hypertrophy (line 231), and LVH line 232

Page 6: from (spelling line 260); alpha-galactosidase A in place of GAL (lines 274 and 276)

Page 8: this paragraph should be clarified, mainly regarding that only about 30% of FD patients are eligible for chaperone therapy. The mention “It acts only on heart function but has no effects on the kidney” should be deleted because i) results on heart were seen regarding LVH, not function; and ii) the results of the 2 pivotal studies conclude that there is a stabilization of kidney function –eGFR with a short follo-up (Germain DP, et al, migalastat vs placebo; Hughes D, et al, migalastat vs enzyme therapy_switch study). More, Lenders M, et al. report in 2020 about decline of kidney function in patients treated with migalastat. This point should be emphasized in this section of the manuscript.

The sentence “…suggesting that also a combined therapy is not entirely effective;” is not clear and should be clarified.

Author Response

Sorriento D, et al. wrote a comprehensive review about Fabry disease cardiovascular phenotype. In a well-written and interesting/complete paper, they analyze the mechanisms of cell damage. I have only minor comments: Since the paper do not include abbreviations section, figures 1 to 3 need a legend (pages 3, 7, and 9, respectively).

Page 1: space between groups and classic (line 39)

Page 2: cells (line 64)

Page 5: ventricular hypertrophy (line 231), and LVH line 232

Page 6: from (spelling line 260); alpha-galactosidase A in place of GAL (lines 274 and 276)

REPLY: We correct the underlined typos.

Page 8: this paragraph should be clarified, mainly regarding that only about 30% of FD patients are eligible for chaperone therapy. The mention “It acts only on heart function but has no effects on the kidney” should be deleted because i) results on heart were seen regarding LVH, not function; and ii) the results of the 2 pivotal studies conclude that there is a stabilization of kidney function –eGFR with a short follo-up (Germain DP, et al, migalastat vs placebo; Hughes D, et al, migalastat vs enzyme therapy_switch study). More, Lenders M, et al. report in 2020 about decline of kidney function in patients treated with migalastat. This point should be emphasized in this section of the manuscript.

The sentence “…suggesting that also a combined therapy is not entirely effective;” is not clear and should be clarified.

REPLY: We corrected according to Reviewer suggestions (lanes 378-384)